# Assessment of environmental variability on malaria transmission in a malaria-endemic rural dry zone locality of Sri Lanka: The wavelet approach

**Rahini Mahendran**[1,©,¤a]*, **Sisira Pathirana**[2,©,¤b]*, **Ilangamage Thilini Sashika Piyatilake**[3], **Shyam Sanjeewa Nishantha Perera**[4], **Manuj Chrishantha Weerasinghe**[5,©]*

**1** Faculty of Medicine, University of Colombo, Colombo, Sri Lanka, **2** Malaria Research Unit, Department of Parasitology, Faculty of Medicine, University of Colombo, Colombo, Sri Lanka, **3** Department of Computational Mathematics, Faculty of Information Technology, University of Moratuwa, Moratuwa, Sri Lanka, **4** Research and Development Center for Mathematical Modeling, Faculty of Science, University of Colombo, Colombo, Sri Lanka, **5** Department of Community Medicine, Faculty of Medicine, University of Colombo, Colombo, Sri Lanka

© These authors contributed equally to this work.
¤a Current address: Smile Train Cleft Centre, Faculty of Dental Sciences, University of Peradeniya, Peradeniya, Sri Lanka
¤b Current address: Institute of Biochemistry, Molecular Biology and Biotechnology, University of Colombo, Colombo, Sri Lanka
* rahini.mahen15@gmail.com (RM); sisira@ibmbb.cmb.ac.lk (SP); manuj@commed.cmb.ac.lk (MCW)

**Data Availability Statement:** Data cannot be made fully available due to third party restrictions. These third party datasets will be obtained upon

## Abstract

Malaria is a global public health concern and its dynamic transmission is still a complex process. Malaria transmission largely depends on various factors, including demography, geography, vector dynamics, parasite reservoir, and climate. The dynamic behaviour of malaria transmission has been explained using various statistical and mathematical methods. Of them, wavelet analysis is a powerful mathematical technique used in analysing rapidly changing time-series to understand disease processes in a more holistic way. The current study is aimed at identifying the pattern of malaria transmission and its variability with environmental factors in Kataragama, a malaria-endemic dry zone locality of Sri Lanka, using a wavelet approach. Monthly environmental data including total rainfall and mean water flow of the "Menik Ganga" river; mean temperature, mean minimum and maximum temperatures and mean relative humidity; and malaria cases in the Kataragama Medical Officer of Health (MOH) area were obtained from the Department of Irrigation, Department of Meteorology and Malaria Research Unit (MRU) of University of Colombo, respectively, for the period 1990 to 2005. Wavelet theory was applied to analyze these monthly time series data. There were two significant periodicities in malaria cases during the period of 1992–1995 and 1999–2000. The cross-wavelet power spectrums revealed an anti-phase correlation of malaria cases with mean temperature, minimum temperature, and water flow of "Menik Ganga" river during the period 1991–1995, while the in-phase correlation with rainfall is noticeable only during 1991–1992. Relative humidity was similarly associated with malaria cases between 1991–1992. It appears that

submitting a formal request to head of the following relevant authorities: i. Malaria Research Unit of University of Colombo (http://med.cmb.ac.lk/) - Malaria cases in Kataragama Medical Officer of Health area ii. Department of Irrigation (http://www.irrigation.gov.lk/) - Data of rainfall and water flow of 'Menik ganga' river iii. Department of Meteorology (http://www.meteo.gov.lk/index.php?lang=en) - Data of temperature such as mean temperature, minimum and maximum temperature (on a nominal charge) Authors confirm that others would be able to access these data in the same manner as the authors and also confirm that they did not have any special access privileges that others would not have.

**Funding:** Sisira Pathirana (SP) was funded by the World Bank, (HETC-QIG-W3 project, Faculty of Medicine, University of Colombo, Sri Lanka) for the field work. The grant number is HETC / QIG / W3 / Medicine - TOR 7. The funders had no roles in study design, data collection and analysis, decision to publish, or preparation of the manuscript.

**Competing interests:** The authors have declared that no competing interests exist.

environmental variables have contributed to a higher incidence of malaria cases in Kataragama in different time periods between 1990 and 2005.

## Introduction

Malaria is a global public health concern. According to the latest World Malaria Report 2018, an estimated 219 million malaria cases, two million more cases compared to 2016, and 435,000 deaths were estimated to have occurred globally in 2017 [1]. As malaria eradication has received due recognition in the global health agenda, the world malaria map is becoming smaller in size. Consistent with WHO goals of reducing malaria incidence and mortality rates and preventing its re-introduction in malaria-free countries, a few countries, including Sri Lanka, have shown major progress towards eliminating malaria.

Sri Lanka was a malaria-endemic country in South Asia and has a long history of malaria pre-dating the colonial period [2]. Malaria control activities in Sri Lanka were established in 1911, and scientifically informed malaria research and interventions began in 1921 [3]. As shown in Fig 1, Sri Lanka suffered a devastating epidemic of malaria in 1934–35 [4]. Following the introduction of DDT (Dichlorodiphenyltrichloroethane) in 1946, malaria was almost eliminated from the country. Only 17 cases were recorded in 1963–64, most of which were imported cases. However, malaria resurfaced in 1967 [5]. Epidemics of malaria were also recorded in 1974–75 [6], 1986–87 and 1991–1995, causing increased malaria incidence and deaths [7]. By early 2000, there was a significant reduction in malaria cases and the country entered the pre-elimination phase in 2008 [8]. It was declared as a malaria-free in 2016. At present, Sri Lanka has seen few imported malaria cases annually during the post-elimination period [9].

Progress towards malaria elimination varies across countries and may depend on the country's health system, availability of funding for malaria control activities, and political and economic commitment. Also, the elimination of malaria has been largely associated with indoor-residual spray, distribution of long-lasting insecticidal nets, the introduction of artemisinin-based combination therapy, passive case detection, major socio-economic development [11], strong community involvement, extensive case detection and ongoing epidemiologic investigation at the local levels [12].

However, the dynamics of malaria transmission are complex processes. Most importantly, the complex dynamics of malaria transmission has been shown to be greatly influenced by environmental factors. Minimum and average temperatures ranging from 15–30˚C provide a vector-friendly environment by increasing the survival of malaria vectors and disease distribution [13, 14]. Increased rainfall creates new mosquito breeding sites whereas drought in wet areas could produce stagnant pools ideal for mosquito breeding sites [15]. The relationship of stream or river water flows with the breeding of malaria vectors and malaria transmission have also been documented [5, 16, 17]. Hence, an in-depth investigation of root causes and factors associated with malaria is crucial on the road to successful malaria elimination.

There are several studies on factors contributing to malaria transmission using mathematical and statistical modelling [18, 19]. These models consist of parameters that vary according to internal factors such as climate and external factors such as demography, geography, and human mobility [20]. Therefore, it is important to identify the effect of these factors on malaria transmission. In a classical approach, this dynamic behaviour is being identified using statistical techniques such as time-series analysis. However, the time-series approach is less reliable for non-stationary epidemiological datasets. Alternative time-series approaches, such as ARIMA models, make data stationary [21]. These ARIMA models have been widely used to

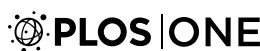

**Fig 1. History of malaria in Sri Lanka from 1911 to 2012 (Reproduced from Abeysinghe et al, 2012 [10]).**

predict periodic and seasonal trends of infectious diseases and are reported to have more predictive power compared to other methods [22]. "Wavelet" analysis is another mathematical tool used for characterizing and estimating dependencies among non-stationary signals [21].

Numerous studies have shown the usefulness of wavelet analysis in analysing the influence of external factors on vector-borne diseases. For example, Cazelles et al [23] found a strong association between the *El Nino* effect and dengue epidemics in Thailand. In a recent study, Abiodun et al [24] revealed a strong annual cycle of malaria incidence with one-year periodicity and a strong relationship between temperature and malaria transmission in KwaZulu-Natal of South Africa. In Sri Lanka, the wavelet approach was useful to identify significant cycles of dengue epidemics and influences of rainfall and maximum temperature on those cycles in urban areas of Colombo [25]. With this background, the use of wavelet analysis in epidemiological investigation of diseases is advantageous when explaining disease transmission cycles under the influence of environmental and other factors.

Kataragama, where the current study was conducted, is in the dry zone of Sri Lanka, the home of endemic malaria. It has been the focus of intense malaria monitoring by MRU, Faculty of Medicine, University of Colombo for almost three decades. Many studies have been conducted in this locality during past 30 years in establishing a relationship between malaria and various other factors. It includes studies aimed at identifying the association between malaria and living and housing conditions [26, 27], the effect of malaria on educational attainment in school children [28], landscapes in relation to water bodies, and deployment of anti-

malaria interventions [27]. However, to the best of our knowledge hardly any published studies have examined the interaction between malaria and environmental variables using wavelet analyses. The objective of this study is, therefore, to identify patterns of malaria transmission and its association with environmental factors for the period from 1990–2005 using wavelet techniques.

## Materials and methods

### Study setting

Ethics approval for this study was obtained from the Ethics Review Committee (EC-12-182) of the Faculty of Medicine, University of Colombo before commencing the study.

Kataragama is a dry lowland coastal plain located in the border of Uva and Southern Provinces of Sri Lanka. It is popular as a pilgrim site and is situated 228 km away from the capital, Colombo. Thirty years ago, the area was a poor locality in the malaria endemic dry zone of Sri Lanka. The study area is approximately 607.01 km$^2$ in extent and surrounded by Yala National Park, a popular wildlife sanctuary. Therefore, human territory is restricted to 12.13km$^2$ of the total area. Based on the 2012 Census of Population and Housing, the total population of the study area was 18,220.

### Malaria and environmental data

Environmental data were obtained from different government departments for the available years. Monthly environmental data, including mean temperature, mean minimum and maximum temperatures, and mean relative humidity from 1990 to 2014, were obtained from the Department of Meteorology Eraminiyaya agro-meteorological field station, the nearest field station to the study area. In addition, monthly total rainfall (1990–2013) and mean water flow rate of the "Menik Ganga" river (1990–2012) were obtained from the Kataragama field station of the Department of Irrigation.

The Field Malaria Research Station of the MRU of the University of Colombo, located in Kataragama, was the main field station in the study area to diagnose, treat, and follow malaria patients. Malaria cases were confirmed through thick blood-smear examination by well-trained and experienced Technical Officers and PhD students under the constant supervision of a Consultant Parasitologist, who was a leading malariologist. Records on malaria cases were properly maintained and regularly reported to the Anti Malaria Campaign. The only other health facility in the area during the study period, Kataragama Divisional Hospital, was a small health establishment. MRU attracted most patients from Kataragama and adjacent villages. Monthly malaria cases from 1990 to 2005 were obtained from MRU.

### Mathematical background to wavelet

**Wavelet transform.** Wavelets [29–31] are mathematical functions that oscillate in a small finite interval. These functions identify localized periodical oscillation in a time series. Wavelet functions consist of two parameters, scaling ($s$) and translation ($u$). With the aid of these two parameters, wavelet functions ($\psi$) extract local time and frequency information from signals.

For a given wavelet $\psi(t)$, a scaled and translated version is defined as

$$\psi_{us} = \frac{1}{\sqrt{s}} \psi\left(\frac{t-u}{s}\right).$$

The wavelet $\psi_{01}(t) = \psi(t)$ is called the *mother wavelet*.

**Continuous wavelet transform.** The Continuous Wavelet Transform (CWT) of a signal $f(t)$ is defined as,

$$W(u,s) = \frac{1}{\sqrt{s}} \int_{-\infty}^{\infty} f(t)\psi^* \left(\frac{t-u}{s}\right) dt, \tag{1}$$

where is $\psi^*(t)$ the complex conjugate of the mother wavelet $\psi(t)$, which is the analysis wavelet function, $u$ is the translation parameter, and $s$ is the scaling parameter. The parameter $u$ is any real number and the parameter $s$ is any positive real number. The mother wavelet $\psi(t)$ is designed to balance between the time domain and frequency domain resolution.

We can clearly see very low-frequency components at large $s$, which makes the width of the mother wavelet expansive, and very high-frequency components at small $s$, which makes the width of the mother wavelet concentrating.

The wavelet function in (1) should satisfy the following two conditions.

$$\int_{-\infty}^{\infty} \psi(t)dt = 0 \text{ and} \tag{2}$$

$$\int_{-\infty}^{\infty} \|\psi(t)\|^2 dt = 1. \tag{3}$$

Considering the Eqs (2) and (3) the original signal $f(t)$ can be reconstructed by means of the inverse CWT and it is given by Calderón's reconstruction formula as

$$f(t) = \frac{1}{C_\psi} \int_{-\infty}^{\infty} \int_{-\infty}^{\infty} W(u,s)\psi_{u;s}(t) \frac{du\,ds}{u^2}, \tag{4}$$

where $C_\psi$ is the admissibility criterion given by the integral

$$C_\psi = \int_{-\infty}^{\infty} \frac{|\hat{\psi}(w)|^2}{w} dw < \infty,$$

and $\hat{\psi}(w)$ is the Fourier transform of $t\omega$.

**Morlet wavelet.** The most widely used mother wavelet in CWT is the Morlet wavelet. It is defined as

$$\psi(t) = \frac{1}{\sqrt[4]{\pi}} \exp(iw_0 t) \exp\left(-\frac{t^2}{2}\right).$$

Here $\omega$ is an adjustable parameter of wave number that allows for accurate signal reconstruction.

**The wavelet power spectrum.** The CWT coefficients are visualized as a scalogram. The wavelet power spectrum [32] of the wavelet transform $W(u,s)$ in (1) is defined as,

$$WPS(u,s) = |W(u,s)|^2. \tag{5}$$

The wavelet power spectrum is averaged over time and compares with the Fourier spectral method. When the average is taken over all times, we can obtain the global wavelet power spectrum, which is defined as

$$GPS(s) = \int_{-\infty}^{\infty} |W(u,s)|^2 du. \tag{6}$$

In the continuous wavelet power spectrum, 5% significant level against red noise is a thick contour. Red and blue colours indicate the stronger and weaker powers, respectively.

**The cross wavelet spectrum.**   Cross wavelet analysis (CWA) is useful to find the frequency correlation between two non-stationary time series. The cross-spectrum of two independent time series $x(t)$ and $y(t)$ is defined as,

$$W_{xy}(u, s) = W_x^*(u, s) W_y(u, s), \tag{7}$$

where $^*$ is the complex conjugate of the wavelet $W_x(u,s)$. The square of the cross power spectrum is normalised using the individual spectrum of each time series. This gives the wavelet coherence, which is defined as,

$$R^2(u, s) = \frac{\left|S\left(W_{xy}(u, s)\right)\right|^2}{S\left(|W_x(u, s)|^2\right) S\left(|W_y(u, s)|^2\right)}, \tag{8}$$

where S is a smoothing operator. It is essential to smooth; if not, all values would be equal to 1. The wavelet coherence value 1 implies a perfect relationship between the two time-series in both time scale, and the value 0 means the two time-series are independent.

In the cross-wavelet spectrum, 5% significant level against red noise is shown as a thick contour. The relative phase relationship between the two signals are indicated using arrows. Arrows pointing right denote in-phase (0˚) relationship between the two signals and arrows pointing left describe anti-phase (180˚) relationship between the two signals. Arrows pointing straight down describe signal 1 leading signal 2 by 90˚ and arrows pointing straight up describe signal 2 leading signal 1 by 90˚.

In the wavelet analysis, the time series data of malaria and environmental factors were mapped into a set of wavelets pertaining to different scales and time instants using the Morlet function [29].

## Analysis of data

In this study, wavelet analysis was performed using the wavelet toolbox in Matlab. Wavelet power spectrums and cross-wavelet spectrums were generated for monthly time-series data of malaria cases and environmental variables assuming a 5% level of significance.

The wavelet approach clearly discloses the changes in different scales (periodic components) of the time-series [21]. It provides the characterization of epidemiological and environmental time-series and possible association between them [23].

Wavelet power spectrums were used to outline dominant modes of malaria and environment time-series and changes of those modes over time along with periodic /cyclic components of malaria cases and environmental variability. Cross-wavelet spectrums were used qualitatively to explore the extent to which malaria cases and environment variables are correlated at certain periods of time.

## Results and discussion

The time series covers 192 months, starting from January 1990 to December 2005. Although environment variables were available from 1990 to 2013/2014, malaria case incidence was markedly decreased from year 2001 and remained zero since 2005. Since a variable having zero values continuously for a considerable time period has no impacts on wavelet and cross-wavelet power spectrums, the wavelet analysis covered only 1990 to 2005. The total number of parasitologically confirmed malaria cases reported during the study period was 24,549 with a

mean of 127.86 cases per month. The number of reported cases per month ranged from 0 to 650.

The environmental data indicated that the monthly mean maximum temperature during the study period ranged from 28.4°C (2002 February) to 34.4°C (2001 May), whereas monthly mean minimum temperature ranged from 19.7°C (1992 February) to 26.3°C (1998 May), respectively. The range of monthly mean relative humidity was 66.5% (1990 August) to 87.5% (1991 October). The highest monthly total rainfall was recorded in 1991 December (641 mm) while the highest monthly mean water flow of "Menik Ganga" river was recorded in 1997 November (63.7 Cumecs).

## Time series and wavelet power spectrums

During the study period, there were few dominant peaks in malaria incidence between 1992 and 1996 and from 1999, as shown in Fig 2A. The rise in malaria incidence documented in Kataragama between 1990 and 2000 and its reduction are consistent with the trend of national malaria morbidity during the study period. According to Anti-Malaria Campaign, Sri Lanka [33], epidemics of malaria were recorded in 1967–69, 1986–87, and in 1990–92. As reported by World Health Organization (WHO) [34], the number of confirmed malaria infections in Sri Lanka increased from 142,294 in 1995 to 265,549 in 1999, then began to decline since 2000, and the reduction was 99.9% from 1999 to 2007. Similarly, the Annual Parasitic Incidence (API) reached a peak of 22.1 per 1,000 in 1999 from 11.9 in 1995 and then declined to less than 1 in 2004 [10].

The malaria time-series data in Fig 2A is non-stationary. Fig 2B shows the wavelet analysis spectrums of malaria cases. The thick contour designates the 5% significance level against red noise. The cone of influence (COI), where edge effects might distort the picture, is shown as a lighter shade. Only the patterns above the cross-hatched region are considered reliable. The color code values are from blue (low values) to red (high values), and the level of significance increases from blue to red. As illustrated in Fig 2B, decomposition of the series indicates time in the x-axis and period (months) in the y-axis. According to Fig 2B, malaria cases show a significant periodicity in the 6th to 12th month in 1992–1995 and in the 2nd to 5th month from 1999 to 2001. In addition, there are shorter periodicities in 1993, 1994, and late 1996, which were less significant compared to other years. These results are consistent with results from Konradsen [35], who reports that malaria epidemics on the island were cyclic from the year 1906 to 1934 and occurred at 5-year intervals. On the other hand, in South Africa, highly significant annual cycles of malaria incidence were found in KwaZulu-Natal province from 1975–2005 [24] and in Mali from 2009–2014 [36]. More recently Anokye et al [37] reported a similar general pattern of malaria infection in Kumasi Metropolis in Ghana from January 2010 to December 2016. Their findings identified the highest number of cases in July and the lowest in January, indicating a cyclic pattern of malaria incidence.

The cycles in Fig 3A over the study period correspond with rainfall between the years 1991–1993 and 1997–2002, and water flow of the "Menik Ganga" river from 1992–1996 and 1997–2000, as shown in Fig 3B. However, using wavelet analysis, Aboidun et al [24] recently ascertained a recurrent cycle with an apparent 1-year period for rainfall in the KwaZulu-Natal province of South Africa from the year 1975–2005, with cyclic patterns observed between 1986–1996 and 2001–2003. The cycles for rainfall in the present study were detected in the 2nd to 7th months in 1991–1993 and the 7th to 12th month in 1997–2002 (Fig 3A), whereas two significant cycles for water flow of "Menik Ganga" river have been detected approximately in 13th to 16th month in 1992–1996 and in the 2nd to 16th month in 1997 and 2000 (Fig 3B).

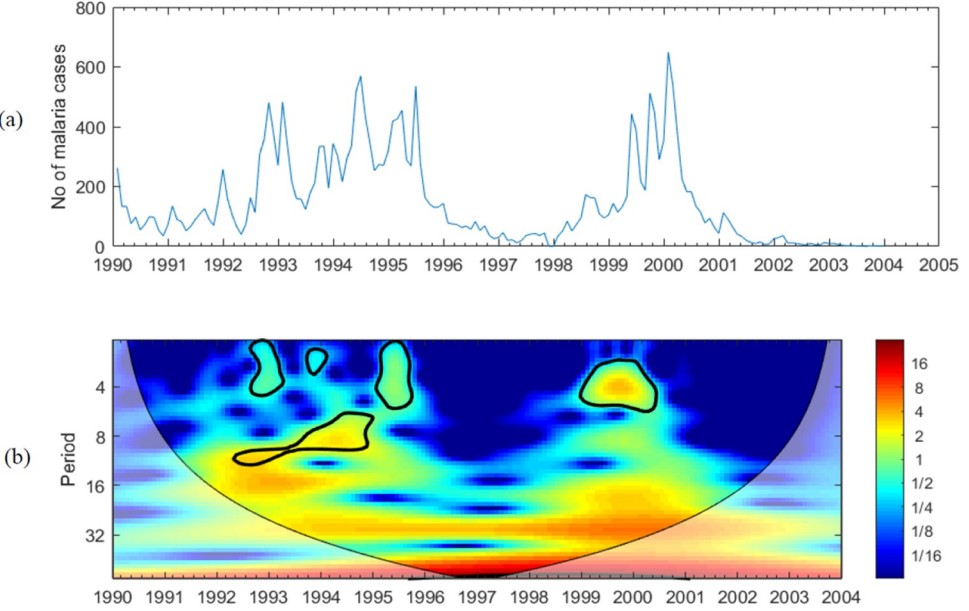

**Fig 2.** (a). Time series of malaria cases from the year 1990–2005 (b). The wavelet power spectrum of malaria cases from the year 1990–2005.

Significantly similar patterns at 13-month are noticeable from 1991 to 2002 for both mean temperature and mean minimum temperature as revealed in the Fig 4A and 4B, whereas mean maximum temperature showed a 13-month cycle in 1993–1995 and 1997–2002 (Fig 4C).

The cycles in Fig 4D over the study period show highly significant overlapping periodicities from the years 2000–2002 and 2001–2003 for relative humidity. In addition, the figure also highlights a recurrent cycle with an apparent 1-year period from 1991–1998, which seems less significant (level of significance increases from blue to red color).

### Cross wavelet spectrums

Figs 5 and 6 show cross wavelet power spectrums of environmental variables and malaria cases over Kataragama from 1990–2005. Faded regions represent the COI and were not considered for analysis. The cross wavelet power spectrums in Fig 5A and 5B reveal correlations between malaria cases and mean temperature and minimum temperature that are noticeably stronger between 1991–1995, while that of rainfall is noticeable only between 1991–1992. Further, the arrows in Fig 5A and 5B indicate the anti-phase relationship of mean temperature and minimum temperature, where a decrease in temperature led to increased malaria incidence during 1991–1995. The arrows in Fig 5C reveal an in-phase relationship of rainfall with malaria cases, in which an increase in rainfall led to a rise in malaria incidence during 1991–1992.

According to previous studies, malaria incidence is closely associated with several climate and environmental factors. Using seasonal autoregressive integrated moving average (SARIMA) forecasting model, Briet et al [38] found a contribution of rainfall seasonality on malaria cases from 1972–2003. Anwar et al [39], using time series analysis, identified significant correlations between malaria and average monthly rainfall and temperature at 0–3 months lag in Afghanistan. Both rainfall and temperature in the KwaZulu-Natal province of South Africa have positively contributed to the transmission of malaria at an average lag of 0 to 120 days [24]. The cross-wavelet analysis of the same study also reveals that the correlation between malaria incidence and rainfall is stronger only during certain years whereas that of temperature is evident through all the years. In addition, similar associations have been documented

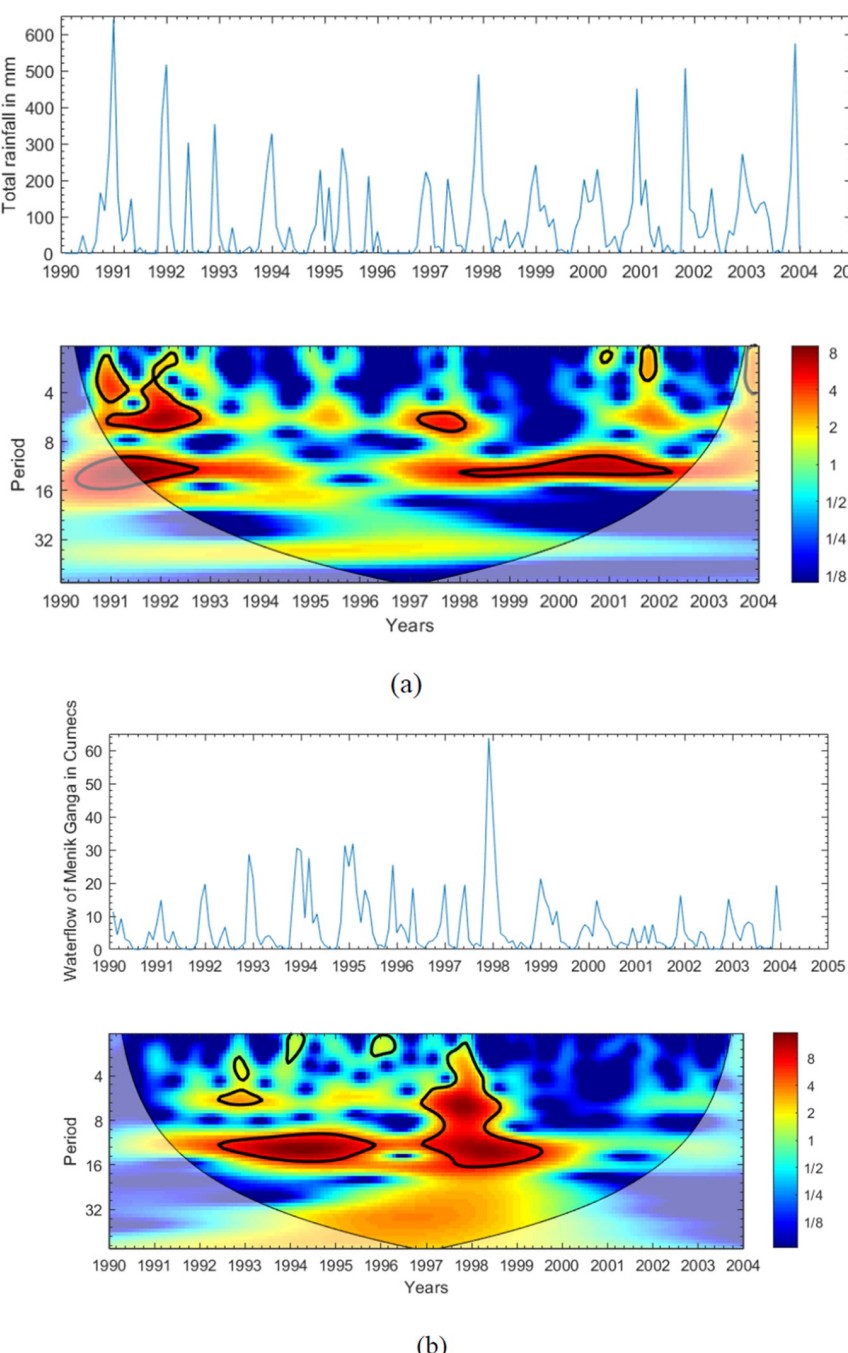

**Fig 3.** (a). The time series and wavelet power spectrums of monthly total rainfall from 1990–2005 (b). The time series and wavelet power spectrums of monthly mean water flow of "Menik Ganga" river from 1990–2005.

for India [40] and China [41], among others. As Cazelles et al [23] reported, dengue epidemics in Thailand are highly related to rainfall and temperature. However, Van der Hoek et al [42] found a weak correlation between monthly rainfall and malaria incidence in Kekirawa, a dry zone locality in North Central province of Sri Lanka, from 1979 to 1995, despite its statistical significance. Similarly, Ostovar et al [43] also found that rainfall is not correlated with malaria infection in South-Eastern Iran.

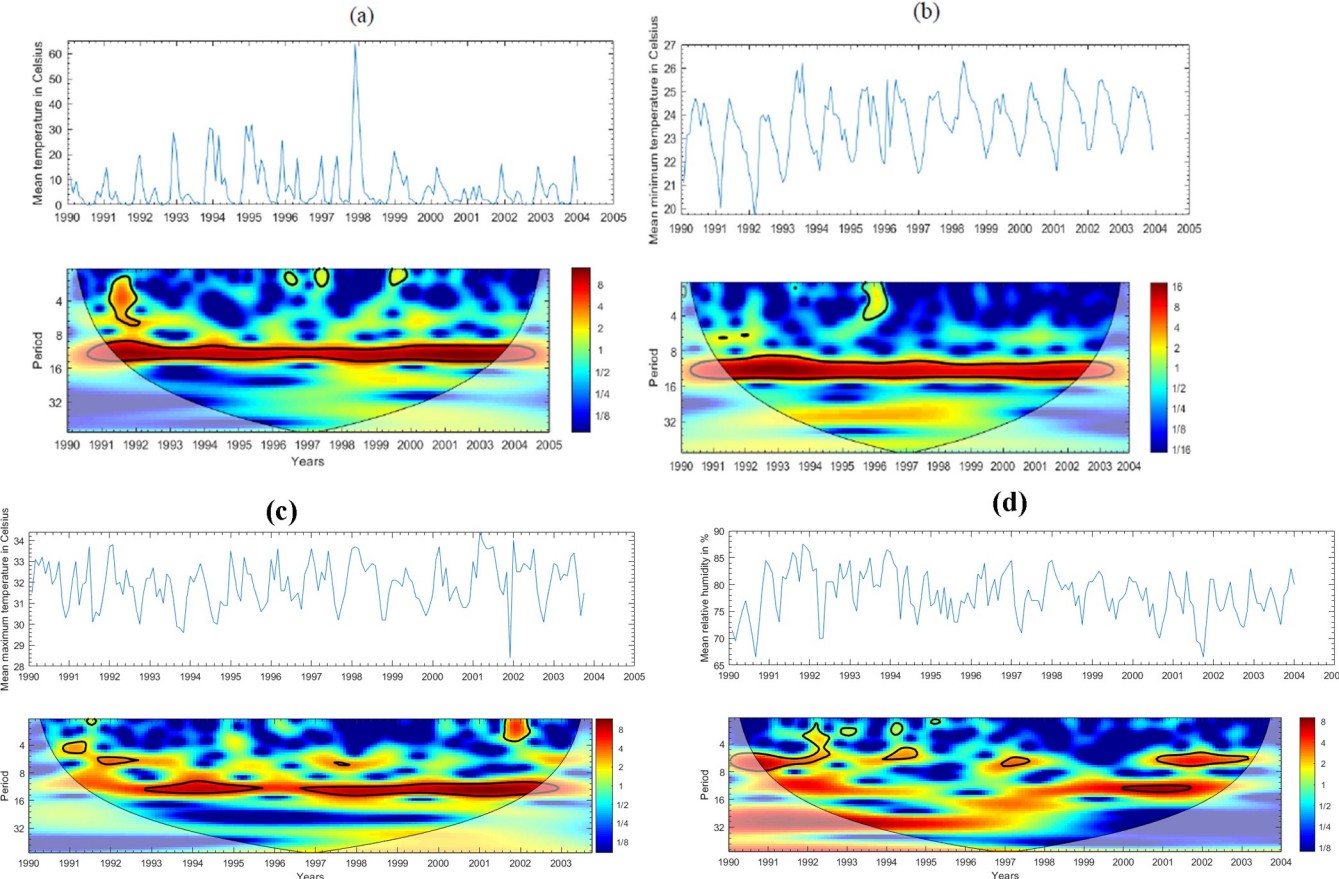

**Fig 4.** (a). The time series and wavelet power spectrums of monthly mean temperature from 1990–2005 (b). The time series and wavelet power spectrums of monthly mean minimum temperature from 1990–2005 (c). The time series and wavelet power spectrums of monthly mean maximum temperature from 1990–2005 (d). The time series and wavelet power spectrums of monthly mean relative humidity from 1990–2005.

The cross wavelet power spectrum in Fig 6A reveals that the water flow of "Menik Ganga" river is associated with malaria cases between months 30 to 70 (1992–1995). Fig 6B indicate a significant relationship between relative humidity and increased malaria cases during the period from 25 to 40 months (1991–1992).

## Conclusions

Malaria is one of the major infectious diseases across the globe, with a history of re-emergence following extended periods of absence, and shows a complex disease transmission. This complex transmission mechanism has not been completely explained by conventional mathematical and statistical models and time-series approaches because epidemiological time-series are complex and strongly non-stationary.

In this study, the wavelet power spectrum was applied to identify significant periodicities of malaria transmission and the influence of environmental factors in those periodicities. Based on findings of this study, we detected an approximately 6–12 and 2–5 month cycle of malaria transmission between the years 1992, 1995, and 1999–2001. The cross wavelet power spectrum analyses showed that the number of malaria cases from 1991 to 1995 was determined by mean temperature, minimum temperature, and water flow of the "Menik Ganga" river; rainfall and relative humidity were more important in 1991–1992. Finally, this paper concludes that temperature, rainfall, water flow of the river, and relative humidity have contributed to higher

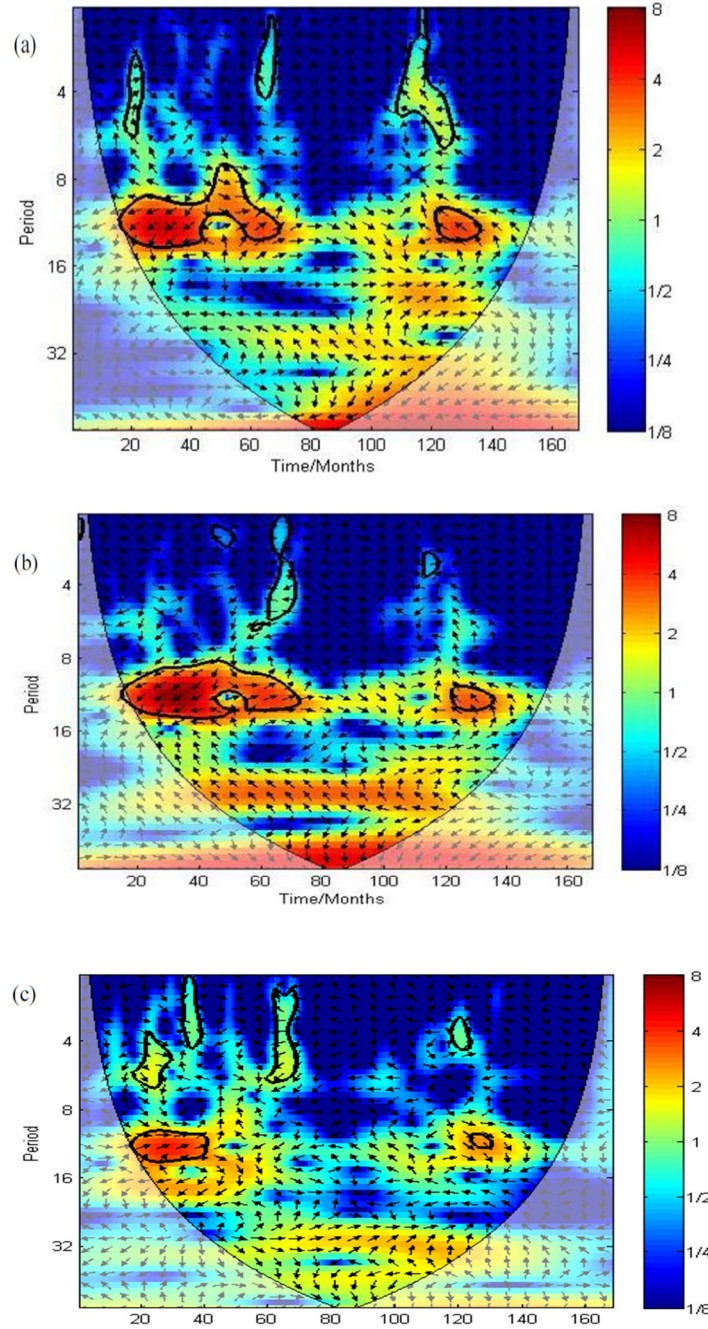

**Fig 5.** (a). Cross wavelet power spectrums of monthly malaria cases with mean temperature in Kataragama from the year 1990–2005 (b). Cross wavelet power spectrums of monthly malaria cases with mean minimum temperature in Kataragama from the year 1990–2005 (c). Cross wavelet power spectrums of monthly malaria cases with total rainfall in Kataragama from the year 1990–2005.

malaria incidence in Kataragama only for the period between 1991 to 1995. Malaria transmission between 1999–2001 was not significantly associated with these environmental factors. It may be possible that factor(s) other than environment may have contributed to the cycle of malaria transmission during this period. Investigators noted drastic changes in the area such as road development and depletion of vegetation in roadsides and gardens that could have

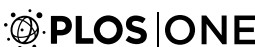

**Fig 6.** (a). Cross wavelet power spectrums of monthly malaria cases with the water flow of "Menik Ganga" river in Kataragama from 1990–2005 (b). Cross wavelet power spectrums of monthly malaria cases with relative humidity in Kataragama from 1990–2005.

reduced mosquito resting places dramatically. However, these factors have not been addressed in the study.

## Acknowledgments

Authors acknowledge Prof. Richard Carter and Prof. Kamini Mendis for conceiving this research idea and late Mr. Dulip Gunasekara for his initial groundwork in this study. Authors

are also thankful to the Department of Irrigation, Department of Meteorology, Anti-Malaria Campaign in Sri Lanka, and the Malaria Research Unit of the University of Colombo in Kataragama for providing secondary data for the period from 1990 to 2014.

Authors acknowledge Prof. Stevan Albert (Graduate School of Public Health, University of Pittsburgh, USA) for copyediting the revised manuscript.

## Author Contributions

**Conceptualization:** Rahini Mahendran, Sisira Pathirana, Manuj Chrishantha Weerasinghe.

**Formal analysis:** Rahini Mahendran.

**Funding acquisition:** Sisira Pathirana.

**Investigation:** Rahini Mahendran.

**Methodology:** Rahini Mahendran, Sisira Pathirana, Manuj Chrishantha Weerasinghe.

**Project administration:** Sisira Pathirana.

**Software:** Ilangamage Thilini Sashika Piyatilake, Shyam Sanjeewa Nishantha Perera.

**Supervision:** Sisira Pathirana, Manuj Chrishantha Weerasinghe.

**Visualization:** Ilangamage Thilini Sashika Piyatilake, Shyam Sanjeewa Nishantha Perera.

**Writing – original draft:** Rahini Mahendran, Ilangamage Thilini Sashika Piyatilake.

**Writing – review & editing:** Sisira Pathirana, Shyam Sanjeewa Nishantha Perera, Manuj Chrishantha Weerasinghe.

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
