## [Decision Letter · Decision Letter 0]

4 Oct 2019

PONE-D-19-18085

Assessment of Environmental Variability on Malaria Transmission in a Malaria-Endemic Rural Dry Zone Locality of Sri Lanka: The Wavelet Approach

PLOS ONE

Dear Ms Mahendran,

Thank you very much for submitting your manuscript "Assessment of Environmental Variability on Malaria Transmission in a Malaria-Endemic Rural Dry Zone Locality of Sri Lanka: The Wavelet Approach" (#PONE-D-19-18085) for review by PLOS ONE. As with all papers submitted to the journal, your manuscript was fully evaluated by academic editor (myself) and by independent peer reviewers. The reviewers appreciated the attention to an important health topic, but they raised substantial concerns about the paper that must be addressed before this manuscript can be accurately assessed for meeting the PLOS ONE criteria. Therefore, if you feel these issues can be adequately addressed, we invite you to submit a revised version of the manuscript that addresses the points raised during the review process. We can’t, of course, promise publication at that time.

We would appreciate receiving your revised manuscript by Nov 18 2019 11:59PM. To enhance the reproducibility of your results, we recommend that if applicable you deposit your laboratory protocols in protocols.io, where a protocol can be assigned its own identifier (DOI) such that it can be cited independently in the future. For instructions see: http://journals.plos.org/plosone/s/submission-guidelines#loc-laboratory-protocols

We look forward to receiving your revised manuscript.

Kind regards,

Abdallah M. Samy, PhD

Academic Editor

PLOS ONE

**Journal Requirements:**

2.Our internal editors have looked over your manuscript and determined that it may be within the scope of our Mathematical Modelling of Infectious Disease Dynamics Call for Papers. The Collection will encompass a diverse range of research articles on using mathematical models to better understand infectious diseases. Additional information can be found on our announcement page: https://collections.plos.org/s/mathematical-disease-dynamics. If you would like your manuscript to be considered for this collection, please let us know in your cover letter and we will ensure that your paper is treated as if you were responding to this call. If you would prefer to remove your manuscript from collection consideration, please specify this in the cover letter.

3. Please note that PLOS ONE has specific guidelines on software sharing (http://journals.plos.org/plosone/s/materials-and-software-sharing#loc-sharing-software) for manuscripts whose main purpose is the description of a new software or software package. In this case, new software must conform to the Open Source Definition (https://opensource.org/docs/osd) and be deposited in an open software archive. Please see http://journals.plos.org/plosone/s/materials-and-software-sharing#loc-depositing-software for more information on depositing your software.

4. Our editorial staff has assessed your submission, and we have concerns about the grammar, usage, and overall readability of the manuscript.  We therefore request that you revise the text to fix the grammatical errors and improve the overall readability of the text before we send it for review. We suggest you have a fluent, preferably native, English-language speaker thoroughly copyedit your manuscript for language usage, spelling, and grammar.

If you do not know anyone who can do this, you may wish to consider employing a professional scientific editing service.  

Whilst you may use any professional scientific editing service of your choice, PLOS has partnered with both American Journal Experts (AJE) and Editage to provide discounted services to PLOS authors. Both organizations have experience helping authors meet PLOS guidelines and can provide language editing, translation, manuscript formatting, and figure formatting to ensure your manuscript meets our submission guidelines. To take advantage of our partnership with AJE, visit the AJE website (http://learn.aje.com/plos/) and enter referral code PLOS15 for a 15% discount off AJE services. To take advantage of our partnership with Editage, visit the Editage website (www.editage.com) and enter referral code PLOSEDIT for a 15% discount off Editage services. If the PLOS editorial team finds any language issues in text that either AJE or Editage has edited, the service provider will re-edit the text for free.

Please note that PLOS ONE does not copyedit accepted manuscripts and that one of our criteria for publication is that articles must be presented in an intelligible fashion and written in clear, correct, and unambiguous English (http://www.plosone.org/static/publication#language). If the language is not sufficiently improved, we may have no choice but to reject the manuscript without review.

**Additional Editor Comments:**

I am wondering that authors reported a Malaria-Endemic Rural Dry Zone; however, Sri Lanka has reported no indigenous cases in the past few years, and in 2016, the World Health Organization certified the country as malaria-free nation. Is there any change in the country status? If Yes, authors should present the detailed picture regarding this point to make it much clearer to the reader.

**Reviewers' comments:**

Reviewer's Responses to Questions

**Comments to the Author**

1. Is the manuscript technically sound, and do the data support the conclusions?

Reviewer #1: Yes

2. Has the statistical analysis been performed appropriately and rigorously? 

Reviewer #1: I Don't Know

3. Have the authors made all data underlying the findings in their manuscript fully available?

Reviewer #1: No

4. Is the manuscript presented in an intelligible fashion and written in standard English?

Reviewer #1: Yes

5. Review Comments to the Author

Reviewer #1: The present article aimed at identifying the environmental variables responsible for malaria trend in a malaria-endemic dry zone locality of Sri Lanka for a 16-years period. This research is relevant and important in the current context of climate change as malaria is very likely to appear or re-emerge in some areas. In that respect, studying the climatic and environmental factors that influence malaria transmission at a regional and local scale will be essential to guide future control and prevention interventions for this vector-borne disease, and this is the focus of this paper. Through their work, Mahendran et al. contribute to improving knowledge on the epidemiology of malaria in Sri Lanka for which associations with environmental variables do not seem to have been widely studied.

Overall, the writing of the manuscript is concise, precise and the methodology is described in an intelligible way. The findings are aligned with the data and the results presented.

However, a few points would benefit from clarification and additional details:

- In the introduction:

o the authors refer to climate as an external factor of malaria transmission. However, environmental drivers such as temperature and rainfall are usually described in the literature as intrinsic factors while extrinsic drivers refer to anthropogenic factors (see: Paaijmans, K. P., & Thomas, M. B. (2011). Health: wealth versus warming. Nature Climate Change, 1(7), 349.)

o I would be more cautious by saying that time-series approach is less reliable for non-stationary datasets as there are procedures for making the data stationary (e.g one or several differentiating in ARIMA models). Plus, ARIMA models have been widely used for the prediction of periodic and seasonal trends in infectious diseases and have been evaluated as having more predictive power compared to other methods (see: Nobre, F. F., Monteiro, A. B. S., Telles, P. R., & Williamson, G. D. (2001). Dynamic linear model and SARIMA: a comparison of their forecasting performance in epidemiology. Statistics in medicine, 20(20), 3051-3069.)

- The authors chose to analyse different environmental drivers including the mean water flow rate of Menik Ganga river but the rational choice behind it is not obvious. Has Menik river been associated with malaria in other studies or is it simply an interest in the water flow? A brief mention in the introduction of the link between stream water flow and malaria would perhaps be welcome, as it was done for temperature and precipitation (possible reference: Konradsen, F., van der Hoek, W., Amerasinghe, F. P., Mutero, C., & Boelee, E. (2004). Engineering and malaria control: learning from the past 100 years. Acta Tropica, 89(2), 99-108.)

- Materials and methods- Malaria and environmental data: it is mentioned that monthly malaria records were maintained properly during the study period (1990 to 2005). However additional information regarding the diagnostic performed would be needed to better understand the data behind those records: what kind of professional performed the diagnostic (clinical officers, medical officers, nurses, and midwives)? Is the recording a systematic and standardized procedure since 1990? Are malaria records confirmed cases? if so, what is the methodology of confirmation (microscopy or RDT)? These two methodologies might present different specificity and sensibility. Do the monthly malaria cases represent prevalence or incidence? Do the data allow to determine between a new case and a recurring one (caused by a treatment failure)?

- The interpretation of the results of the wavelet power spectrums is not that easy to follow for non-experts in wavelet analysis. What does a cycle of mean maximum temperature or water flow represent? Is it a particular fluctuation of temperature or water flow? Does the detection of a cycle in water flow in the 13th to 16th month from 1992-1996 mean that three cycle of three months occurred during this 60-months period? Just a few additional details on the cycle and their month of detection could help improve the understanding of results.

- The conclusion of the analysis is that environmental factors had contributed to malaria incidence (or prevalence) only for the period 1991-1995 but a cycle of malaria transmission was also detected during 1999-2001. Does it mean that factors other than environment have influenced the transmission or that there might not be enough statistical power to detect an effect? If environmental factors are only correlated to the transmission in 1991-1995, what conclusion/recommendation can be drawn from these results for the purpose of surveillance, early warning/forecasting?

Furthermore, few repetitions, grammatical and punctuation errors are presents in the manuscript, which would require a new reading and some rewording.

Few examples:

- In the abstract: There are evidences to say that malaria transmission is largely depends -> transmission largely depends

- In the intro:

o The objective of this study is therefore, aimed at identifying the patterns … -> ... is, therefore, aimed at identifying patterns

o As malaria eradication been given due recognition in the global health agenda, world malaria map is becoming smaller in size. -> the world map

- In the metho : Itis essential to smooth… -> it is essential

- In cross wavelet spectrum: In addition, similar associations were documented for India [35] and China [36] -> have been documented for India...

- ...

6. PLOS authors have the option to publish the peer review history of their article (what does this mean?). If published, this will include your full peer review and any attached files.

Reviewer #1: No

---

## [Author Response · Author response to Decision Letter 0]

16 Jan 2020

Our responses to the reviewer’s comments are given in a separate document which was uploaded to the online portal and the manuscript was revised accordingly.

---

## [Editor Report · Decision Letter 1]

21 Jan 2020

Assessment of Environmental Variability on Malaria Transmission in a Malaria-Endemic Rural Dry Zone Locality of Sri Lanka: The Wavelet Approach

PONE-D-19-18085R1

Dear Dr. Pathirana,

We are pleased to inform you that your manuscript has been judged scientifically suitable for publication and will be formally accepted for publication once it complies with all outstanding technical requirements.

With kind regards,

Abdallah M. Samy, PhD

Academic Editor

PLOS ONE

---

## [Editor Report · Acceptance letter]

3 Feb 2020

PONE-D-19-18085R1 

Assessment of Environmental Variability on Malaria Transmission in a Malaria-Endemic Rural Dry Zone Locality of Sri Lanka: The Wavelet Approach 

Dear Dr. Pathirana:

I am pleased to inform you that your manuscript has been deemed suitable for publication in PLOS ONE. Congratulations! Your manuscript is now with our production department. 

With kind regards,

on behalf of

Dr. Abdallah M. Samy 

Academic Editor

PLOS ONE